# *Achromobacter* spp. in a Cohort of Non-Selected Pre- and Post-Lung Transplant Recipients

**DOI:** 10.3390/pathogens11020181

**Published:** 2022-01-28

**Authors:** Cornelia Geisler Crone, Omid Rezahosseini, Hans Henrik Lawaetz Schultz, Tavs Qvist, Helle Krogh Johansen, Susanne Dam Nielsen, Michael Perch

**Affiliations:** 1Centre of Excellence for Health, Immunity and Infections (CHIP), Copenhagen University Hospital, Rigshospitalet, 2100 Copenhagen, Denmark; 2Department of Infectious Diseases, Copenhagen University Hospital, Rigshospitalet, 2100 Copenhagen, Denmark; Omid.rezahosseini@regionh.dk (O.R.); tavs.qvist@regionh.dk (T.Q.); sdn@dadlnet.dk (S.D.N.); 3Department of Cardiology, Section for Lung Transplantation, Copenhagen University Hospital, Rigshospitalet, 2100 Copenhagen, Denmark; Hans.Henrik.Lawaetz.Schultz.01@regionh.dk (H.H.L.S.); Michael.Perch@regionh.dk (M.P.); 4Department of Clinical Microbiology, Copenhagen University Hospital, Rigshospitalet, 2100 Copenhagen, Denmark; hkj@biosustain.dtu.dk; 5Department of Clinical Medicine, Faculty of Health and Medical Sciences, University of Copenhagen, 2200 Copenhagen, Denmark; 6The Novo Nordisk Foundation Center for Biosustainability, Technical University of Denmark, 2800 Lyngby, Denmark

**Keywords:** *Achromobacter*, lung transplantation, solid organ transplantation, cystic fibrosis, incidence, mortality

## Abstract

*Achromobacter* is an opportunistic pathogen that mainly causes chronic lung infections in cystic fibrosis (CF) patients and is associated with increased mortality. Little is known about *Achromobacter* spp. in the lung transplant recipient (LTXr) population. We aimed at describing rates of *Achromobacter* spp. infection in LTXr prior to, in relation to, and after transplantation, as well as all-cause mortality proportion in infected and uninfected LTXr. We included 288 adult LTXr who underwent lung transplantation (LTX) between 1 January 2010 and 31 December 2019 in Denmark. Bronchoalveolar lavage was performed at regular intervals starting two weeks after transplantation. Positive cultures of *Achromobacter* spp. were identified in nationwide microbiology registries, and infections were categorized as persistent or transient, according to the proportion of positive cultures. A total of 11 of the 288 LTXr had transient (*n* = 7) or persistent (*n* = 4) *Achromobacter* spp. infection after LTX; CF was the underlying disease in 9 out of 11 LTXr. Three out of the four patients, with persistent infection after LTX, also had persistent infection before LTX. The cumulative incidence of the first episode of infection one year after LTX was 3.8% (95% CI: 1.6–6.0). The incidence rates of transient and persistent infection in the first year after LTX were 27 (12–53) and 15 (5–37) per 1000 person-years of follow-up, respectively. The all-cause mortality proportion one year after LTX was 27% in the *Achromobacter* spp. infected patients and 12% in the uninfected patients (*p* = 0.114). *Achromobacter* spp. mainly affected LTXr with CF as the underlying disease and was rare in non-CF LTXr. Larger studies are needed to assess long-term outcomes of *Achromobacter* spp. in LTXr.

## 1. Introduction

*Achromobacter* is a genus of non-fermentative aerobic Gram-negative rods, mainly found in the environment in water and soil. Opportunistic disease in humans is well described and typically manifests as chronic respiratory infections in persons with chronic lung disease, such as cystic fibrosis (CF) or in immunocompromised patients [1,2,3,4]. Elderly patients with comorbidities and patients with malignancies are also known to be susceptible. Extra-pulmonary manifestations such as sepsis, bacteremia, urine-, and catheter or device infections are seen [1,3,5,6].

Infections with *Achromobacter* spp. are difficult to treat because of intrinsic and acquired antibiotic resistance. In addition, there is no standard treatment protocol [7], although it has been suggested that early treatment with inhaled and oral antibiotics postpones time to the next culture of *Achromobacter* spp. in CF patients [8].

In addition, *Achromobacter* form biofilm in the respiratory and conductive zones of the lung, the latter of which is poorly vascularized, leads to lower antibiotic availability when treatment is attempted either by oral intake or intravenously [8,9]. Due to this, infections in the respiratory tract often become persistent. These lead to continuous inflammation and are associated with poor long-term outcomes in infected CF patients [10,11], where an excess decline in lung function of −1.6% annually (95% CI −2.2 to −0.90) has been reported. This leads to premature death or lung transplantation 16.8 years after initial colonization [12].

In elderly patients with hospital-acquired *Achromobacter* pneumonia, 30-day mortality has been reported as high as 33%, despite antibiotic therapy [4]. *A. xylosoxidans* has been found as the causative agent in outbreaks of hospital-acquired infection, possibly associated with the bacilli’s capability of forming biofilms, creating reservoirs in water systems or even contaminating solutions [4,7,13,14].

Lung transplantation (LTX) is a last resort treatment for patients with end-stage lung disease, such as CF, emphysema, or pulmonary fibrosis. All lung transplant recipients (LTXr) are treated with immunosuppressives following transplantation to avoid graft rejection. This iatrogenic immunosuppression leads to increased risk of infections [15]. In most transplant centers, LTXr are frequently hospitalized as part of the follow-up program, where routine examinations include bronchoscopy with bronchoalveolar lavage (BAL).

*Achromobacter* in CF patients undergoing LTX have previously been described and early re-infection has been reported [16,17,18]. However, these studies also found that *Achromobacter* re-infection did not reduce survival in CF patients after LTX [16,17,18].

The general LTXr population has several risk factors for developing *Achromobacter* spp. infection including prior severe lung disease, immunosuppression, and frequent contacts to the healthcare system with invasive procedures. However, information about *Achromobacter* spp. infections in the general LTXr population is lacking [19]. We aimed to determine the incidence of *Achromobacter* spp. prior to, in relation to, and after LTX in LTXr, regardless of underlying disease. We also aimed to describe manifestation of infection, time to infection/re-infection, persistency of infection, and all-cause mortality proportion in *Achromobacter* infected LTXr.

## 2. Results

We identified 298 patients who had received a LTX during the inclusion period. We excluded three patients due to age < 18 years and seven patients, where the LTX in the inclusion period was not the first LTX.

### 2.1. Baseline Characteristics

Eleven out of the 288 LTXr who were included had transient or persistent *Achromobacter* spp. infection after transplantation. The median age at transplantation was significantly lower in LTXr with *Achromobacter* spp. infection compared to the uninfected group (33 vs. 53 years, *p* = 0.006). This was due to the high proportion of CF patients in the *Achromobacter* spp. group having 82% with CF as the underlying disease, compared to 12% in the uninfected group (Table 1). Most patients underwent double lung transplantation in both groups.

### 2.2. *Achromobacter* spp. Pre-Transplantation

Within the year prior to transplantation, 155 of the 288 included LTXr had at least one microbial culture performed, and 9 out of the 155 had at least one culture-positive for *Achromobacter* spp. Six out of the nine fulfilled the criteria of having persistent infection within the year prior to LTX. The remaining three had transient infection in the year leading up to LTX and had never had persistent infection prior to LTX.

All nine patients had CF as the underlying disease and were positive in respiratory tract specimens. The median (IQR) time difference from LTX to the last positive culture prior to transplantation was −6 (−19 to −4) days.

### 2.3. *Achromobacter* spp. in Relation to Transplantation

All 288 LTXr had at least one culture performed in relation to the time of transplantation, and five had at least one culture-positive for *Achromobacter* spp. All these patients had persistent infections in the year leading up to LTX. The *Achromobacter* spp. were cultured in blood in all of the five patients (two peripheral blood, two central venous catheters, and one port-a-cath). Four out of the five patients had simultaneous positive cultures from airway secretions taken related to LTX.

### 2.4. *Achromobacter* spp. after Transplantation

During the first year after transplantation, 11 LTXr had one or more cultures positive for *Achromobacter* spp. The cumulative incidence of the first episode of infection at day 365 post-LTX was 3.8% (95% CI: 1.6–6.0) (Figure 1).

Of the 11 infected patients, 9 (82%) were CF patients and 2 (22%) were non-CF patients (Table 2). The median time to the first *Achromobacter* spp. positive culture after transplantation was 10 days (IQR 5–33). In 7 out of 11 (64%) *A. xylosoxidans* was cultured after LTX. Three patients (27%) had positive cultures with *A. rhulandii*. One patient had positive cultures with *Achromobacter species* without further identification (Table 2). Whole genome sequencing was applied to pre- and post-LTX isolates in seven patients (patient no. 1, 3, 5–9, Table 2). Patients 5 and 6 had different species identified post-LTX compared to the one pre-LTX. Patients 1, 3, 7–9 had identical strains of *Achromobacter xyl**osoxidans* identified pre- and post-LTX, where isolates that differed <5000 SNVs in the core genome were considered identical [20]. *Achromobacter rhulandii* was found in patients 2, 4, and 6. The *A. rhulandii* is also called the Danish epidemic strain as described by Ridderberg et al. in 2011 and 2012 [21,22].

With 259 person-years of follow up, the incidence rate of persistent and transient infection in the first year post-LTX was 15 (5–37) and 27 (12–53) per 1000 person-years of follow up, respectively.

### 2.5. Characteristics of Infected Non-CF Patients

Two non-CF patients were transiently positive early after transplantation and were not positive prior to LTX (patients 10 and 11, Table 2). Patient 10, with emphysema as an underlying disease, had one positive culture of *A. xylosoxidans* in a sputum sample at day 37. The patient did not have clinical signs of respiratory infection and no antibiotic treatment was prescribed. The patient was alive at end of the study.

Patient 11, with pulmonary fibrosis as an underlying disease, had a positive *Achromobacter species* culture of BAL at day 42. The patient did not have respiratory symptoms and was not treated with antibiotics at that point. *Achromobacter species* was found again in a BAL culture at day 90. Due to no signs of clinical infection, antibiotic treatment was not initiated. A total of 23 respiratory specimens were cultured during the first year after LTX and only the two were positive. At end of follow-up, the patient was alive and had had no further positive cultures of *Achromobacter* spp.

### 2.6. Characteristics of Infected CF Patients

Nine CF patients were infected in the first year after LTX whereas eight were also positive prior to LTX (Table 2). The one patient, who was not infected prior to transplantation, had transient infection after LTXr and died on day 12 (patient 4, details in Section A.1). Six (67%) had persistent infections and three (33%) had transient infections during the first year after LTX. Of the five transiently infected patients, two had only one positive respiratory specimen during the first year after LTX (patients 2 and 9). Of the nine patients infected prior to LTX, one patient cleared the infection related to LTX (details in Section A.2).

### 2.7. Mortality

The crude all-cause mortality proportion within the first-year post-LTX was 3 (27%) out of 11 in LTXr with *Achromobacter* spp. compared to 33 (12%) out of 277 uninfected (*p* = 0.114). In the CF population during the first year, the crude all-cause mortality proportion was 6 (15%) out of all 41 CF patients and 3 (33%) out of the 9 *Achromobacter* spp. infected CF patients compared to 3 (9.5%) out of the 32 uninfected CF patients (*p* = 0.108).

The cause of death in the four patients surviving >1-year post-LTX was chronic graft rejection (*N* = 3) and bacterial infection (*N* = 1, patient 3) (Table 2). Patient 3 was chronically infected with *Achromobacter* spp. at time of death and died due to respiratory failure from superinfection with Staphylococcal pneumonia. The cause of death of patient 7, who died on day 179 post-LTX was “Infection,” which referred to cytomegalovirus infection

## 3. Discussions

In our cohort of 288 LTXr, *Achromobacter* spp. was mainly found in patients with CF and was rare in non-CF patients. *Achromobacter* spp. was transiently found in two non-CF patients who had no clinical signs of infection.

Previous studies on multi-drug resistant Gram-negative bacilli in solid organ transplant recipients (SOT) have been performed but only limited data are available on *Achromobacter* spp. infection in these patients. In a recent study of 143 SOT, *A. xylosoxidans* was reported in one patient, resulting in a prevalence of 0.4%. However, no information of organ transplant type or underlying disease was reported [23]. In our study, *Achromobacter* spp. was cultured in two non-CF patients with emphysema and pulmonary fibrosis as the underlying diseases. One patient was positive in a single sputum sample during hospital admission and had no symptoms of respiratory infection. This positive culture could be caused by a swap of samples from patients, leading to incorrect patient identification, or due to contamination while handling the specimen. The other patient, also positive during hospital admission, had two positive cultures from the lower respiratory tract, reducing the likelihood of contamination. Despite immunosuppression due to recent transplantation, this patient did not seem susceptible to infection with *Achromobacter* spp. The patients had no signs of respiratory infection and no need for antibiotic treatment. In a review of 20 case studies of *Achromobacter* respiratory infection, the most common chronic conditions, after CF, was malignancies (*N* = 15) and non-CF bronchiectasis (*N* = 3) [3]. The review did not report any cases of emphysema, pulmonary fibrosis patients, or LTXr. *Achromobacter* pneumonia has recently been reported in patients with similar conditions to our non-CF patients in a study of 15 cases of hospital-acquired *Achromobacter* pneumonia from China, including patients with chronic obstructive pulmonary disease and chronic bronchitis [4]. However, the two non-CF patients did not seem susceptible for *Achromobacter* spp. infection in our study.

In the CF patients infected prior to LTX, we found re-infection occurred early after transplantation, which has also been reported previously in our center in a study on CF patients following LTX [16] with an overlap in patients included in our study (see Appendix B). The study reported a median of 23 days to re-colonization of four common Gram-negative pathogens including *Achromobacter* [16].

In our study, most CF patients, with persistent infection in the year leading up to LTX, were also positive at the time of transplantation, often in both respiratory specimens and blood cultures from catheters. Whether these findings represent contamination or actual bacteremia is uncertain. High mortality of *Achromobacter* bacteremia has been reported with 23% all-cause 30-day mortality [5]. The patients with positive cultures related to the transplant surgery did not have high early mortality rates in our study. It is possible that these positive findings represent contamination of skin or catheters during admission to the surgical ward. We found 9 CF patients infected with *Achromobacter* prior to LTX and 8 (89%) of them were re-infected after transplantation. Previous studies have reported lower proportions of recurrence. A study by Nolley et al. included 26 CF patients infected prior to LTX and isolated *Achromobacter* after transplantation in 12 (46%) patients [18]. Within the first year after LTX, Lobo et al. reported recurrence in 9 out of 9 (100%) CF patients with pan-resistant *Achromobacter* and 5 out of 15 (33%) with multi-drug resistant *Achromobacter* [17]. Our relatively high rate of recurrence may relate to antibiotic resistance patterns of the *Achromobacter* spp. involved or to insufficient eradication treatment in the window of opportunity after LTX. Bacterial seeding from the paranasal sinus could also affect the high rate of re-infection since the sinuses often serve as a reservoir for these chronic infective pathogens [24,25]. Eradication of sinus infection, with sinus surgery related to LTX, is also practiced in some patients in our center, but results regarding the effect of this invasive procedure are ambiguous [26,27,28,29].

When evaluating the all-cause mortality during the first year after LTX, we found a not statistically significant, higher proportion in the infected group compared to the uninfected. The small numbers limit the validity of this finding and must be interpreted thereafter. Previous larger studies have not been able to detect an effect on mortality in *Achromobacter* infected patients. In a study of 186 LTXr with CF, Lobo et al. identified 24 patients infected with *Achromobacter* and found no difference in the 5 year-survival rates when comparing infected with uninfected [17]. Another study of 89 LTXr with CF, identified 26 patients infected with *Achromobacter* prior to LTX and found no significant difference in the one year (0.84% vs. 0.94%) and three year survival rates (0.68% vs. 0.84%) [18]. In LTXr, infections are believed to be a possible trigger of chronic rejection in the form of chronic lung allograft dysfunction (CLAD) [30]. It is likely that *Achromobacter* infection also serves as such a trigger, affecting the long-term outcomes in timelines beyond the existing literature. We found that three out of four infected patients, surviving more than one year after LTX, had chronic graft rejection as the cause of death. Larger studies with a focus on long-term outcomes, including pulmonary exacerbations, lung function decline, CLAD, and late mortality are warranted.

To our knowledge, this is the first study describing *Achromobacter* in a cohort of LTXr. Our study is strengthened by the access to national registries of all microbiological results collected in the clinical practice and linked to each patient through the Danish Civil Registration System. Our study also has several limitations including its retrospective design and a small number of *Achromobacter* spp. infected patients. Patients with even a single positive sputum culture after LTX were categorized as transient infection, although the positive culture could have unclear clinical significance. Furthermore, a degree of sampling or detection bias could confound our results. Sampling in CF patients could be more frequent compared to sampling in non-CF patients. However, all LTXr were followed with routine microbiological cultures during the first year after LTX including bronchoscopies. Lack of sampling prior to the first two-week routine BAL could affect the time-to-infection reported and should be kept in mind. In addition, we did not have information about *Achromobacter* spp. colonization in the upper airways, which could be an important source for reinfection. We also did not have information about other co-existing microorganisms with poor outcomes such as *Burkholderia cepacia* complex and *non-tuberculous mycobacteria*. Due to overlap between the data sources, some of the CF patients that we included were also included in previous reports (Appendix B).

## 4. Materials and Methods

We included all adult first-time LTXr, who underwent lung transplantation between 1 January 2010 and 31 December 2019 at the Danish National Lung Transplantation Centre, Rigshospitalet, University of Copenhagen. We excluded patients under age 18 and patients where the LTX performed in the inclusion period was not their first LTX.

### 4.1. Data Sources

Data regarding transplantation and patient characteristics were retrieved from the national lung transplantation database. Data regarding microbiological cultures and death were collected from nationwide registries through the Centre of Excellence for Personalized Medicine of Infectious Complications in Immune Deficiency (PERSIMUNE) data warehouse [31]. In PERSIMUNE, data were prospectively collected and merged as part of the routine care of the transplant recipients. Data from several national registries and clinical databases, such as the national Danish Microbiology Database (MiBa), are merged in the PERSIMUNE data repository. MiBa contains all microbiological data from patients who were examined in both general practices or in Denmark hospitals since 2010 [32].

### 4.2. Definitions

#### 4.2.1. *Achromobacter* spp. Infections

*Achromobacter* spp. infections were categorized according to the type of specimen and the proportion of positive cultures. Respiratory tract infection was defined as a positive culture of *Achromobacter* spp. in BAL fluid, sputum, or tracheal/endo-laryngeal specimens.

Persistent infection was defined as a patient with ≥50% of all respiratory tract specimens positive for *Achromobacter* spp. (with ≥4 specimens collected), during a 365-day period leading up to or after LTX, according to modified Leeds criteria [33]. Patients had to be positive in specimens taken over a minimum period of three months to qualify as having persistent infection. Transient infection was defined as having ≥1 positive specimen of any type and not meeting the definition for persistent infection [34]. Infection related to transplantation was defined as any specimens positive for *Achromobacter* spp. sampled at day −1, 0, or 1 after transplantation.

#### 4.2.2. Cultures

Microbiological specimens were cultured according to the standard of care, e.g., Gram-stain and microscopy. Culture was performed on selective and diagnostic agar including antibiotic susceptibility testing. Before 2011 the isolate of *Achromobacter* spp. was identified using analytical profile index with 20 miniature biochemical tests for identification of Gram-negative non-Enterobacteriaceae (API 20NE) (bioMérieux,Marcy-l’SEtoile, France). After 2011 a matrix-assisted laser desorption ionization-time of flight (MALDI-TOF) (Bruker, Billerica, MA, USA) technique was used for identification [20]. Since 2020 Whole genome sequencing (WGS) for typing and identification has been done on all first *Achromobacter* spp. since MALDI-TOF typing is not accurate for *Achromobacter* spp. level typing [8,20].

#### 4.2.3. Immunosuppression and Antimicrobial Prophylaxis

All LTXr received induction therapy consisting of methylprednisolone and thymoglobuline followed by maintenance therapy with a calcineurin-inhibitor, prednisolone, and an antiproliferative agent (azathioprine or Mycophenolate Mofetil). Two antibacterial agents were administered systemically for the first couple of weeks after transplantation, with meropenem and ciprofloxacin being the standard choice. In some patients with known bacterial infections, treatment was targeted according to susceptibility. During the first three months after transplantation, antiviral prophylaxis consisted of valganciclovir. Antifungal prophylaxis consisted of universal voriconazole (2010–2016) or targeted posaconazole and amphotericin B inhalations (2016–2020). All patients received lifelong sulfamethoxazol and trimethoprim as *Pneumocystis jirovecii* (PCP) prophylaxis.

#### 4.2.4. Microbiological Surveillance

All patients were followed with a routine surveillance program consisting of bronchoscopy with BAL sampling after transplantation at weeks 2, 4, 6, and 12 and at months 6, 12, 18, and 24 after transplantation. All BAL were examined with microscopy, culturing, antibiotic susceptibility testing, and typing. Additional bronchoscopy with BAL or other respiratory tract sampling was performed on clinical indication.

#### 4.2.5. Incidence of *Achromobacter* spp. Infections after Transplantation

Included LTXr were followed from day two after transplantation to a positive *Achromobacter* spp. culture, re-transplantation, death, end of the first-year post-transplantation, or 28 February 2021, whichever came first. The last date of inclusion was 31 December 2019 to allow a minimum of one-year follow-up time for all included LTXr.

#### 4.2.6. Cause of Death

The cause of death was obtained by the Classification of Death Causes after Transplantation (CLASS) method, where all deceased LTXr were evaluated by clinicians in a standardized procedure, as previously described [35].

### 4.3. Statistics

Proportions and continuous data were reported as percentages and medians with interquartile ranges (IQR). Fisher’s exact test was used to test the frequency distributions. Mann–Whitney U test was used to compare the differences in medians. The 95% confidence intervals (CI) of incidence rate were calculated using Byar’s approximation to the Poisson distribution. Estimates of the cumulative incidence of the first episode of infection were calculated using the Aalen–Johansen estimator. Statistical analyses were performed using R software version 4.1 (RStudio Version 1.2.5001) and *p*-values ≤ 0.05 were considered statistically significant.

## 5. Conclusions

We found that *Achromobacter* spp. infection was not common in our LTXr cohort and was mainly found in CF patients. *Achromobacter* spp. was cultured in two non-CF patients, who did not show significant clinical symptoms of *Achromobacter* spp. infection. A high proportion of CF patients were re-infected after transplantation. Re-infection occurred early, but not all CF patients became persistently infected. No statistically significant difference in all-cause mortality proportions the first year after LTX was found between infected and uninfected patients. Larger studies are needed to assess long-term outcomes of *Achromobacter* spp. infection in LTXr.

## Figures and Tables

**Figure 1 pathogens-11-00181-f001:**
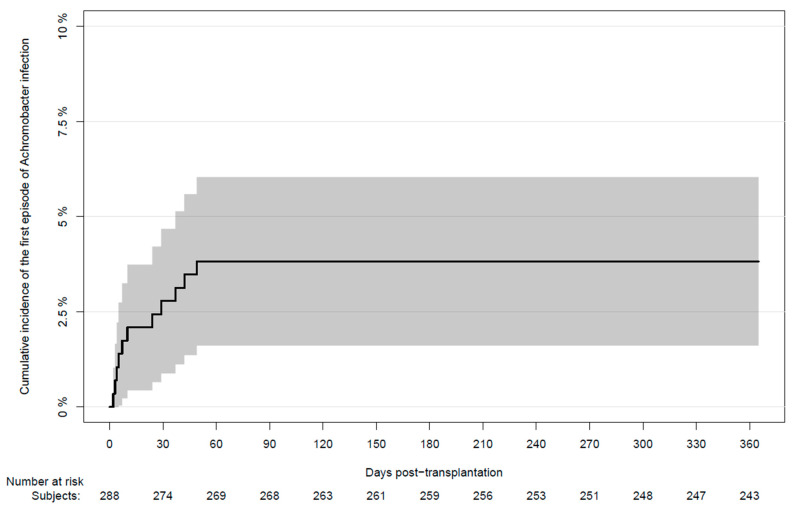
Cumulative incidence of the first positive *Achromobacter* spp. culture in lung transplant recipients during the first year after transplantation. Bronchoalveolar lavage was performed regularly from week two after transplantation and onwards. Additional respiratory sampling was performed on the indication.

**Table 1 pathogens-11-00181-t001:** Baseline demographics of lung transplant recipients grouped according to *Achromobacter* spp. infection status after transplantation.

	Non-Achromobacter (*n* = 277)	*Achromobacter* spp. (*n* = 11)	Total (*n* = 288)	*p*-Value
Age at transplantation, median (IQR)	53 (46–59)	33 (29–45)	53 (44–58)	0.006
Male gender, *n* (%)	144 (52)	6 (55)	150 (52)	1.0
Double lung transplantation, *n* (%)	249 (90)	11 (100)	260 (90)	0.540
Underlying disease, *n* (%)				
Cystic fibrosis	32 (12)	9 (82)	41 (14)	
Emphysema	136 (49)	1 (9.1)	137 (48)	
Pulmonary fibrosis	100 (36)	1 (9.1)	101 (35)	
Pulmonary hypertension	9 (3.2)	0 (0.0)	9 (3.1)	<0.001

IQR = interquartile range, *n* = number of patients.

**Table 2 pathogens-11-00181-t002:** Characteristics of lung transplant recipients with positive *Achromobacter* spp. culture after transplantation.

Patient No	Underlying Disease	*Achromobacter* spp. Pre-LTX	Persistent Infection Pre-LTX	*Achromobacter* spp. in Relation to LTX	Days to Positive Culture	Species	Persistent Infection	Dead	Days to Death	Cause of Death
1	CF	Yes	Yes	Yes	2	A.X ^1,2^	No	Yes	1560	Rejection ^3^
2	CF	Yes	Yes	Yes	7	A.R ^2^	No	Yes	3284	Rejection ^3^
3	CF	Yes	No	No	49	A.X ^1,2^	No	Yes	1950	Bacterial infection
4	CF	No	No	No	5	A.R ^2^	No	Yes	12	Primary graft failure ^4^
5	CF	Yes	No	No	24	A.S ^1^, A.X ^2^	Yes	No		
6	CF	Yes	Yes	Yes	4	A.X ^1^, A.R ^2^	Yes	Yes	2096	Rejection ^3^
7	CF	Yes	Yes	No	3	A.X ^1,2^	Yes	Yes	179	Infection
8	CF	Yes	Yes	Yes	10	A.X ^1,2^	Yes	Yes	349	Cardiac or vascular failure ^5^
9	CF	Yes	No	No	29	A.X ^1,2^	No	No		
10	Emphysema	No	No	No	37	A.X	No	No		
11	Fibrosis	No	No	No	42	A.S	No	No		

CF = Cystic fibrosis, Fibrosis = Pulmonary Fibrosis, pre-LTX = prior to lung transplantation, *Achromobacter* spp. in relation to LTX = positive culture at day −1, 0, or 1 from transplantation, and Days to positive culture = days to first positive *Achromobacter* culture from time of transplantation. Dead = dead status by end of follow-up, A.X= Achromobacter xylosoxidans, A.S = Achromobacter species unknown, and A.R = Achromobacter rhulandii. (1) Identified pre-LTX by whole genome sequencing, (2) Identified after LTX by whole genome sequencing, (3) Graft rejection, chronic, (4) Graft failure, primary non-function, (5) Organ failure or dysfunction (not due to graft rejection, graft failure, GvHD, or infection), Cardiac or vascular.

## Data Availability

Data used in this study are available on request from the corresponding author. The data cannot be made public due to Danish legislation.

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
