# Peer review of "Achromobacter spp. in a Cohort of Non-Selected Pre- and Post-Lung Transplant Recipients"

_pathogens, 2022, doi:10.3390/pathogens11020181_

Round 1

Reviewer 1 Report

General comments:

The authors describe Achromomobacter in lung transplant (LTX) recipients, before and after LTX, and for CF and non-CF patients. They also assess mortality one year post LTX based on positive cultures for Achromobacter. The purpose is good and the topic is of clinical interest, specifically in CF.

Depending on the number of positive cultures (modified Leeds criteria) patients were categorised as having transient or chronic infection. The definitions of “transient” and “chronic” are described in detail further down in the methods section. I lack a discussion of “infection”: should one single positive sample in an asymptomatic patient post LTX (with no positive samples pre LTX) be called transient infection? Is it rather just a positive sputum culture with unclear significance? How to justify the grouping “infected” for this individual in the survival analyses?

It is important to differ between re-colonisation/re-infection (in CF patients), occasional positive cultures in an asymptomatic patient post-transplant and chronic infection requiring antibiotic treatment. Reinfection from the upper airways is discussed in the discussion section but not considered in the data analyses.

Results are given in numbers and percentage. When it comes to small numbers (below 10) I prefer the actual numbers instead of percent. For the abstract alternative writing could then be:  “Eleven of the 288 LTXr had transient (n=7) or persistent (n=4) Achromobacter spp. infection after LTX and CF was the underlying disease in 9/11 LTXr. Persistent reinfection occurred in 4 individuals with CF with Achromobacter infection pre-transplant, all 4 with chronic infection before LTX? (Or did anyone with chronic infection post LTX have transient infection preLTX? This is not clear from the presented data).

The authors calculated the all-cause mortality proportion one year after LTX in Achromobacter spp. infected and uninfected patients, respectively, and based on presence of Achromobacter spp in sputum or BAL samples, occasionally or repeatedly and with or without treatment. The time (in days) until first positive sample and the number of positive samples depends on the standardised sampling methods. BAL was performed on regular intervals in all patients which is a strength of the study but also affect the number of days until a positive cultures.  If no cultures were taken prior to the scheduled two-week bronchoscopy then no samples could be positive before this time point.

Sub-grouping for CF and non-CF was done but only individuals with CF had chronic Achromobacter before LTX. Further sub-analyses had been of interest but not done due to small numbers.

No significance in mortality between the groups with or without Achromobacter was found. The choice of, and the limitations of the grouping method should be discussed. For example, no information is given of other co-existing microorganisms with prognostic poorer outcome such as Burkholderia cepacia complex and non-tuberculous mycobacteria.

Row 171,  2.7, is described one patient dying after more than one year post LTX from bacterial infection ”covering Achromobacter spp. or other bacterial infection”. The bacteria causing the death should be specified. The patient (no 3) had no Achromobacter post LTX and is not anticipated to die from Achromobacter infection. For patient no 7 who had chronic Achromobacter and died from infection it is of interest to know the causative microorganism, was it from severe Achromobacter infection the patient died?

Some comments

Abstract:

row 21: “Lung transplant recipients are at risk for Achromomobacter sspp. infection but little is known of the disease in this population”. I suggest to remove this sentence as evidence is lacking regarding the risk for A. infection. Also, to in the same sentence write of “infection” and of “disease” implies that the Achromobacter infections described are severe -a disease- while in fact one single positive sample in an asymptomatic patient is defined as infection.  

Bacterial sampling methods for cultures should be described in the abstract ("BAL on regular intervals at .......", or "BAL on regular intervals starting 2 weeks post LTX").

Table 1 The Achromobacter group has lower median age, which is due to CF diagnose. This could be commented in results.

Table 2 I suggest that you mark patients with chronic A infection pre LTX with an asterix (are all 4 with persistent infection post LTX also chronic pre LTX?). The title of column 5 should be explained in text below, what does “in relation to LTX “ mean?

Figure1. Describe the sampling method in text below the figure.

The data and topic are overlapping with the data in reference 16. Seven/9 CF patients in the present manuscript was included and analysed for recurrence of bacteria post LTX in reference 16.

Reviewer 2 Report

Pathogens 1546217

Achromobacter spp. in a Cohort of Non-Selected Pre- And Post- 2 Lung Transplant Recipients

Cornelia Geisler Crone 1,2 ◊ *, Omid Rezahosseini 2 ◊; Hans Henrik Lawaetz Schultz 3, Tavs Qvist 2, Helle Krogh Jo- 4 hansen 4,5,6, Susanne Dam Nielsen 2,5 and Michael Perch

This manuscript describes Achromobacter spp. infections in pre- and post-lung transplant patients with CF, and in those with other conditions. The authors concluded that infections with this pathogen mainly affected CF patients and were rare in non-CF patients. The study data were clearly presented, and the study itself provided valuable information about these organisms and their role in CF lung disease.

Minor comments:

It is slightly confusing to have the abstract sections (background, methods, results and conclusion) divided by numbers (1-4). It would be better to remove the numbers.

Line 81: determining, rather than determing

Line 125: Please re-phrase “and one patient did not have the type of Achromobacter species identified”.

Line 286: Please use “isolate” rather than “strain”

Line 291: The authors mention that from 2020 onwards whole genome sequencing was used to identify and type some of these isolates. It would be interesting to know whether any sets of pre and post isolates from individual patients had been typed, and whether they represented the same strain. In addition, was there any information about the strain type of the A. ruhlandii isolates? Were any of them the Danish epidemic strain described by Ridderberg et al., 2011?
